# Experimental inoculation of pigs with porcine parainfluenza virus 1 revealed pathological manifestations in the upper respiratory tract

Marianne Viuf Agerlin[1☯], Kasper Pedersen[1☯], Mathias Romar[1], Marta Canuti[1], Timm Harder[2], Nicole Bakkegård Goecke[1], Henrik Elvang Jensen[1], Lars Erik Larsen[1], Pia Ryt-Hansen[1], Charlotte Kristensen[1]*

1 Department of Veterinary and Animal Sciences, University of Copenhagen, Frederiksberg C, Denmark,
2 Institute of Diagnostic Virology, Friedrich-Loeffler-Institut, Island of Riems, Greifswald, Germany

☯ These authors contributed equally to this work
* chark@sund.ku.dk

## Abstract

Several pathogens are known to affect the respiratory tract of pigs resulting in decreased health and welfare. Porcine parainfluenza virus 1 (PPIV1) and swine orthopneumovirus (SOV) have been identified as novel viruses in pigs. The pathogenicity of PPIV1 has been investigated experimentally by one research group, whereas SOV is yet to be studied. In this experimental trial, two groups of eight pigs were inoculated with a European isolate of PPIV1 or a pool of SOV RNA-positive clinical nasal swab material, and one group of four pigs with culture medium only (negative controls). Nasal swab samples were regularly collected to investigate viral RNA shedding, tissue samples for viral RNA and histopathological examinations, and blood samples to investigate seroconversion. All SOV inoculated pigs tested negative for SOV at 4 days post inoculation (DPI) and therefore, four of these pigs were transferred to the group with the PPIV1-infected pigs to assess direct-contact transmission. At DPI 4, two control and four pigs from each of the PPIV1 and SOV groups were euthanized and necropsied. The remaining pigs were euthanized at 14 DPI. No clinical signs, except for nasal discharge, were observed in any of the pigs. PPIV1 RNA shedding was observed from DPI 2–11 with peaks between DPI 4 and 7, and PPIV1 was transmitted horizontally to all direct-contact pigs. The highest viral RNA load was detected in the upper respiratory tract, i.e., nose, upper and lower trachea compared to the lower respiratory tract, i.e., bronchioles, and alveoli. Generally, a chronic tracheitis at 4 DPI, developing into chronic, erosive tracheitis at 14 DPI was observed in the PPIV1 groups and was supported by in situ detection of PPIV1 by RNAscope. Three pigs also developed mild, bronchointerstitial pneumonia at 14 DPI. All PPIV pigs euthanized at 14 DPI seroconverted. In conclusion, these results showed that PPIV1 is a primary porcine respiratory pathogen that causes breakage of the tracheal epithelial barrier and therefore can predispose to secondary infections.

**Data availability statement:** The sequence of the whole PRV1 genome used for inoculum has been uploaded to GenBank with accession number PV767405. All other relevant data are within the manuscript and its Supporting Information files.

**Funding:** The project was partly founded by the CoVetLab project PorParEU (2024-2025 to L.EL) and the Danish Swine Levy Foundation (2024 to L.E.L). The funders had no role in study design, data collection and analysis, decision to publish, or preparation of the manuscript.

**Competing interests:** The authors have declared that no competing interests exist.

SOV's role as a porcine respiratory pathogen remains unknown, since no successful infection was established and it was not isolated in cells either.

## Author summary

Respiratory diseases in pigs impair porcine health and welfare and are often caused by a complex interaction between pathogens. We investigated the pathogenesis of two recently discovered viruses: porcine parainfluenza virus 1 (PPIV1) and swine orthopneumovirus (SOV). An experimental model in weaner pigs showed that PPIV1 is capable of efficient replication, horizontal transmission, and caused pathological manifestations mainly in the upper respiratory tract. The lesions consisted of chronic, erosive tracheitis in all PPIV1-infected pigs, resulting in a breakage of the tracheal epithelial barrier, and a few pigs developed bronchointerstitial pneumonia. These findings confirm that PPIV1 should be considered a primary porcine respiratory pathogen. SOV could not be isolated in cells, and using pooled nasal swabs as inoculum failed to establish infection, and the reasons why remain unclear.

## Introduction

In intensive swine production, high stocking density makes respiratory pathogens a threat to swine health, also affecting negatively welfare, growth and, thus the sustainability of swine production [1–4]. Porcine parainfluenza virus 1 (PPIV1) is an enveloped single-stranded negative-sense RNA (ssRNA) virus in the family *Paramyxoviridae,* genus *Respirovirus* and species *Respirovirus suis* (formerly known as *Porcine respirovirus type 1*) [5,6]. The PPIV1 genome consists of 15,298 nucleotides (nts) and encompasses six open reading frames (ORFs) coding for: nucleoprotein (N), phosphoprotein (with C/V editing), matrix (M) protein, fusion (F) protein, hemagglutinin-neuraminidase (HN) protein, and a large polymerase (L) [5,7–9].

PPIV1 was first identified among pigs in China in 2013 [10], and was isolated in cell culture in 2016 [11]. It is widely present among conventional pigs globally. A passive surveillance study in Germany found PPIV1 in 76/123 herds [12], in Poland it was detected in 23/30 herds [13], and in Hungary and Slovakia in 11/24 herds [14]. Lower rates of detections have also been reported in Germany and the Netherlands, where 11/34 herds tested positive for PPIV1 [15]. In USA, 365/842 of routine diagnostic samples tested positive [16]. Similar high detection rates were reported in Chile [17] and Italy [18]. In Denmark, PPIV1 was detected in 22/125 herds in an active surveillance project (submitted from pigs with a variety of symptoms such as coughing and diarrhea) performed in 2022 and in 33/176 herds in a passive surveillance project in 2023 [19].

The pathogenesis and pathogenicity of the European PPIV1 lineage have yet to be elucidated, but experimental studies in pigs with the American PPIV1 lineage have

been performed [5,7,8]. No clinical signs were observed in a PPIV1 experimental infection of caesarian derived-colostrum deprived (CDCD) pigs, whereas mild coughing was present later during the infection (6–16 days post inoculation (DPI)) in PPIV1 experimentally infected conventional pigs [5]. These studies also revealed that PPIV1 mainly affects epithelial cells in the upper respiratory tract (URT) of pigs, causing mild transient tracheitis and mild bronchointerstitial pneumonia in some of the experimentally infected pigs [5,7,8].

Swine orthopneumovirus (SOV) is another virus recently discovered among pigs. This virus was discovered in the United States in 2016 [20], and since then, it has been detected in France [21], Spain [22], Germany (12), Sweden [23], and Asia [24]. SOV, which has not yet received an official species designation, belongs to the family *Pneumoviridae,* and genus *Orthopneumonovirus*, which includes human- and bovine respiratory syncytial virus and pneumonia virus of mice (HRSV and BRSV, respectively, and PVM) [6,20,21]. Like PPIV1, SOV is an enveloped, single-stranded negative-sense RNA virus [20], and the pathogenesis and pathogenicity of this virus have not yet been assessed. Additionally, this virus has not been isolated in cell culture.

The aims of the present study were to genetically characterize an European clade I isolate of PPIV1 and to assess the pathogenesis and horizontal transmission of this virus by experimental inoculation of conventional weaned pigs. Further-more, another group of pigs was inoculated with nasal swab material from pigs that tested positive for SOV RNA in an attempt to also study the pathogenesis of this virus. The study also allowed collection of PPIV1-positive tissue samples for distribution to colleague laboratories throughout Europe, enhancing collaborations regarding the development and valida-tion of robust diagnostic detection methods of PPIV1 in the frame of the CoVetLab project PorParEU (2024–2025).

## Materials and methods

### Ethics statement

The pig experiment was performed at the animal experimental facilities at University of Copenhagen from May to June 2024. The experiment was approved by the Danish Animal Experimentation council under license number 2024-15-0201-01620 and performed in biosafety level 2 conditions.

### Preparation of inoculum

Two cell lines, LLC-MK2 (ATCC, CCL-7) and Calu-3 (ATCC, HTB-55) were maintained at 37 °C in a humidified 5% $CO_2$ incubator and grown in cultivation media consisting of Eagle's Minimal Essential Media (MEM) (Gibco, Thermo Fisher Scientific, Waltham, MA, USA) supplemented with 100 U/ml penicillin-streptomycin (Invitrogen, Thermo Fisher Scientific), 2mM L-Glutamine (Sigma-Aldrich, St. Louis, MO, USA), and 10% fetal bovine serum (Gibco). Calu-3 cells were addition-ally supplemented with 1mM Sodium pyruvate (Gibco). The PPIV1 isolate GER/2022AI03675/2022 was propagated in LLC-MK2 cells for three passages. The PPIV1 isolate was inoculated into confluent monolayers of LLC-MK2 in either 48 flat-bottom Nunclon Delta Surface Plates (Thermo Fisher Scientific) or T75 tissue culture flasks (Thermo Fisher Scientific). After incubation for one hour and without removal of the inoculum, serum-free cultivation media supplemented with 2 µg/ml L-1-tosylamido-2-phenylethyl chloromethyl ketone treated trypsin (Sigma-Aldrich) was added. The supernatant was harvested after 72 – 96 hours, the presence of viral nucleic acid was estimated by reverse-transcriptase real-time PCR (RT-qPCR) [12].

For the attempt to propagate SOV, four SOV RT-qPCR positive nasal swab samples, collected from pigs of 4 – 6 weeks of age obtained from two Danish sow herds (herd size of ~800 sows) included in a longitudinal field study [25], were pooled and filtered through a 0.45 µm membrane filter (Lab Logistics Group GmbH, Hamb, Meckenheim, Germany). One hundred µL of the pooled sample and 100 µL cultivation-media with only 2% fetal bovine serum (Thermo Fisher Scientific) was inoculated into LLC-MK2 cells in 48 flat-bottom Nunclon Delta Surface Plates (Thermo Fisher Scientific) in triplicate. An identical setup for SOV isolation was performed in Calu-3 cells. After incubation for two hours, an additional 800 µL of

cultivation media supplemented with 2% fetal bovine serum was added. Six days after inoculation, the presence of SOV RNA was investigated using RT-qPCR analysis [12], and seven DPI, a second passage with the same setup as described above was performed on both cell lines. Samples with Ct-values < 30 were considered true positives.

For the final SOV inoculum, the resuspension fluid of 44 SOV-positive nasal swab samples (18 mL in total) obtained from the same two Danish sow herds were all pooled and filtered as described above. The presence of SOV RNA in the SOV inoculum was investigated using RT-qPCR analysis [12], and additionally, it was evaluated for the presence of other relevant respiratory pathogens using a high-throughput RT-qPCR analysis (both methods described in further detail below).

## Study design

Twenty 4-week-old Norsvin Landsvin pigs were imported from a herd at NMBU (Norwegian University of Life Science) in Norway to obtain pigs without compromising agents, such as swine influenza A virus (IAV) and porcine reproductive and respiratory syndrome virus type 1 and 2 (PRRSV-1 and PRRSV-2). The pigs were confirmed negative for shedding of PPIV1 and SOV by RT-qPCR nucleic acid detection on nasal swabs prior to inoculation, and serum was confirmed negative for antibodies against PPIV1 by ELISA [26]. The presence of SOV antibodies was not assessed due to the lack of an SOV-antibody detection assay. The pigs were stratified by weight and randomly allocated into three groups (stratified randomization): PPIV1 ($n=8$), SOV ($n=8$), and negative controls ($n=4$) (Sex and bodyweight information are available in S1 Table). To reduce any respiratory bacteriological interference, the pigs were treated with 2.5 mg/kg tulathromycin at -7 DPI.

At 0 DPI, the pigs were anesthetized by an intramuscular injection (0.1 mL/kg) by a mixture of Zoletil 50 Vet (without solvent) mixed with 6.25 mL Rompun (20 mg/mL), 2.50 mL Torbugesic (butorphanol tartrate) (10 mg/mL), and 1.25 mL Ketaminol (100 mg/mL). After induction of anesthesia, all pigs were intranasally inoculated in each nostril by MAD nasal intranasal mucosal atomization devices. PPIV1 (a German isolate, GER/2022AI03675/2022-(SK398), accession number PV767405) inoculum consisted of 1.5 mL of $1.19 \times 10^4$ 50% tissue culture infectious dose per mL ($TCID_{50}$/mL, back-titrated) in MEM, SOV inoculum included 1.5 mL of pooled SOV from nasal swab samples (cycle threshold (Ct)-value of 20.72), and the control inoculum was 1.5 mL MEM. Nasal swab samples were collected from all pigs at 0, 1, 2, 4, 7, 9, 11, and 14 DPI. Rectal temperatures were measured at 0, 1, 2, 4, 7, and 14 DPI. Blood samples and body weight measurements were collected at 0, 4, and 14 DPI. Clinical signs, including nasal discharge, coughing, and sneezing, were recorded when observed. At 4 DPI, four PPIV1, four SOV, and two control pigs were euthanized for macro- and microscopic evaluation and virological investigation of tissues. At 4 DPI, no shedding of SOV was detected in any of the SOV-inoculated pigs, and therefore, the last four SOV-inoculated pigs were transferred to the stable of the four PPIV1 pigs as "recipients" to investigate horizontal PPIV1 transmission (0 days post contact, DPC). The study design is illustrated in Fig 1.

The nasal swab samples were collected with flexible sterile rayon dryswabs (Medical Wire, Corsham, UK) inserted into the ventral meatus of each nostril of the pig, turned 360 degrees, and placed into 2 mL Sigma Virocult medium (Medical Wire). Blood was collected from the *vena jugularis* in a BD Vacutainer and 10 mL serum tubes (BD, Stockholm, Sweden).

## Necropsy and histopathology

In anesthesia, the pigs were euthanized by an intracardiac injection of pentobarbital (400 mg/ml, ≥ 0.25 ml/kg). At necropsy, the lungs were photographed, and lung lesions were sketched manually. The percentage of affected lung tissue on the dorsal and ventral surface was measured from the lung photographs using the lung sketches and the area tool in Adobe Acrobat Reader. Pictures from pig number 550 were excluded from the analysis due to poor quality. Samples from the nose, upper and lower trachea, and three different lung sections were collected from each pig (Fig 1). From all pigs euthanized DPI 14, samples from *Ln. tracheobronchialis cranialis* were also collected. Tissue specimens for virus quantification were stored at −80°C until analysis. A post-mortem bronchoalveolar lavage (BAL) was performed by cutting the right lung from the right main bronchus and administering 5 ml phosphate-buffered saline (PBS) into the bronchi

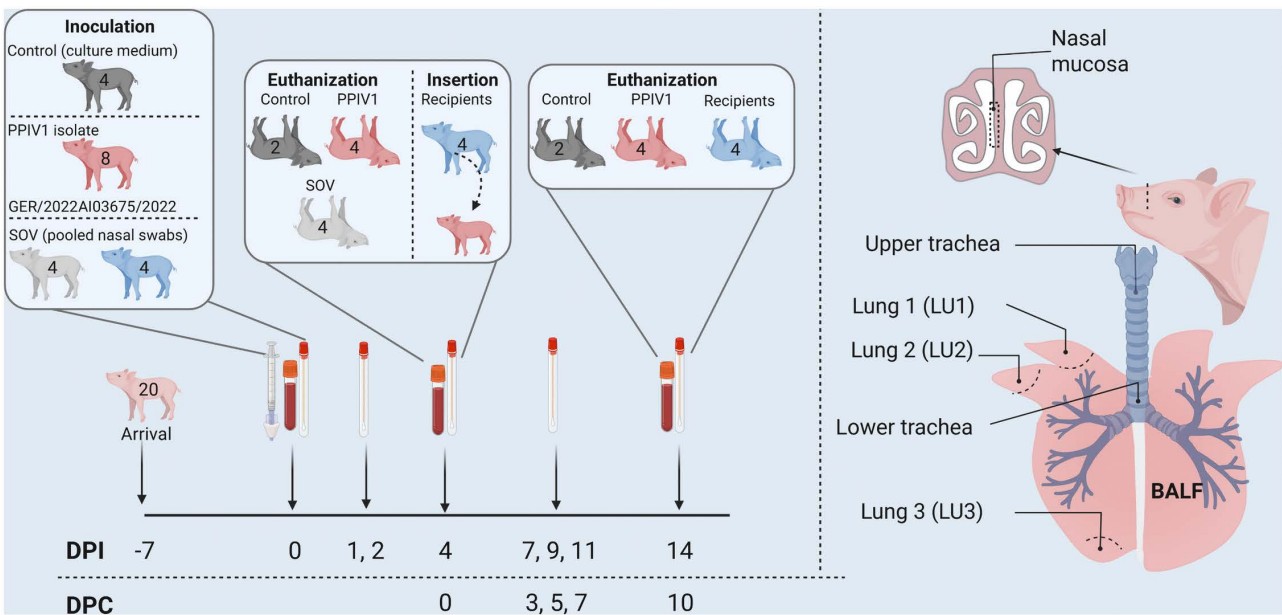

**Fig 1. Overview of the study design.** All pigs were clinically monitored from admission to the experiment facilities one week before and until 14 days post inoculation (DPI). Control, PPIV1 and SOV pigs were inoculated at 0 DPI. Serum samples were collected at 0, 4, and 14 DPI, and nasal swabs were collected at day 0, 1, 2, 4, 7, 9, 11 and 14 DPI. Two controls, four PPIV1 and four SOV pigs (grey-colored) were euthanized at 4 DPI. PPIV1-naïve recipient pigs (blue-colored from the SOV group) were mixed with the PPIV1 inoculated pigs at 4 DPI (0 days post contact, DPC). Tissue samples and bronchoalveolar lavage fluid (BALF) were collected during necropsy at 4 DPI and 14 at DPI/10 DPC, whereas the remaining pigs were euthanized. Created in BioRender. Kristensen, **C.** (2026) https://BioRender.com/6uyj8hn.

using a syringe, massaging the lungs, and the collected BAL fluid (BALF) was stored at −80°C. Specimens for histopathology were fixed in 10% neutral-buffered formalin for five days, embedded in paraffin wax (formalin-fixed and paraffin-embedded, FFPE), sliced into 3–4 µm sections, and stained with hematoxylin and eosin (H&E). FFPE tissue samples of the nose were not evaluated due to variable quality.

## Tissue culture infectious dose 50 (TCID50)

To determine the viral titer of the inoculum, a TCID50 assay with RT-qPCR readout was performed. LLC-MK2 cells were seeded in 24-well plates and cultured until reaching confluency. The virus inoculum was serially diluted tenfold in serum-free cultivation media supplemented with 1 µg/ml TPCK-treated trypsin (Sigma-Aldrich). Prior to inoculation, cells were washed with PBS, and 200 µl of each virus dilution was added to the wells in five replicates. Plates were incubated for one hour at 37 °C in a humidified 5% $CO_2$ incubator. Following the incubation, the inoculum was removed, cells were washed with MEM, and 1 mL of serum-free cultivation media supplemented with 1 µg/ml TPCK-treated trypsin was added to each well. The cells were incubated for three days under the same conditions. To assess viral replication, a sample was collected before and after incubation and analyzed by RT-qPCR. The TCID50 was calculated using the Reed-Muench method.

## Cultivation of viral samples

To assess if the viral samples collected during the animal study were infectious, the samples were cultivated in LLC-MK2 cells. A total of four nasal swab samples, five BALF samples, one serum sample, and twelve tissue samples were diluted and filtered through a 0.45 µm membrane filter. The samples were inoculated into LLC-MK2 cells as described above, and the presence of viral nucleic acid was monitored by RT-qPCR. Cultivation was performed in two independent attempts.

## Antibodies against PPIV1

Serum samples collected from PPIV1-infected, control (0 and 14 DPI), and recipient pigs (0 and 10 DPC) were tested for the presence of anti- PPIV1 antibodies. First, the blood was centrifuged at 1550×g for 10 minutes at 4°C, and sera were stored at -20°C until analysis. Samples were tested for antibodies towards PPIV1 by Iowa State University Veterinary Diagnostic Laboratory (ISU VDL) with a whole virus (WV) indirect ELISA previously described [26]. The true negative ratio (TNR) for a cut-off of 0.14 is evaluated to be 99% (CI 97–100%).

## RNA extraction and reverse transcription real-time PCR

The nasal swab samples were stored at -80°C until RNA extraction. Each nasal swab sample was vortexed for 10 sec, centrifuged for 3 min at 9651×g, and 200 µL of the supernatant was mixed with 400 µL RTL-buffer (QIAGEN, Hilden, Germany) containing 2-mercaptoethanol (ME) (Sigma-Aldrich). In total, 70 mg of tissue were mixed with 1400 µL of RLT buffer, and lysed in a TissueLyser LT (QIAGEN) by bead beating for 3 min at 30 Hz, centrifuged for 3 min at 9651×g, and 600 µL of the supernatant was used for RNA extraction. The RNA was extracted from the nasal swab and tissue samples using the RNeasy mini kit (QIAGEN) automated on the QIAcube Connect extraction robot (QIAGEN) according to the instructions from the manufacturer. For the blood samples collected 4 DPI, 140 µL serum was used for RNA extraction using the Viral kit (QIAGEN) and automated on the QIAcube Connect extraction robot (QIAGEN) according to the instructions from the manufacturer.

For nucleic acid RT-qPCR detection of PPIV1, already published primers targeting the fusion (F) gene were used. The primer-probe-mix consisted of 0.4 µL PRespiV-FF (100 pmol/µl), 0.4 µL PRespiV-F-R (100 pmol/µl), 0.1 µL PRespiV-F-FAM (100 pmol/µl), and 3.1 µL RNase free water [12]. For SOV detection, already published primers targeting the nucleoprotein (NP) gene were used, whereas a previously published probe (Pneumo-NP-FAM) was modified with a FAM in the 5' end and BHQ3 in the 3' end (FAM-CTG GGC TGC CTG ACA ATC GGA GGC-BHQ1) [12]. The SOV primer-probe mix consisted of 0.4 µL Pneumo-NP-F (100 pmol/µl), 4.0 µL Pneumo-NP-R (100 pmol/µl), 0.1 µL Pneumo-NP-FAM (100 pmol/µl), and 3.1 µL RNase free water. In both RT-qPCR assays, the primer-probe-mix was combined with 2.5 µL RNase free water, 12.5 µL RT-PCR buffer, 1 µL of 25x RT-PCR enzyme mix from the AgPath-ID One-step RT-PCR kit (Applied Biosystems, Thermo Fisher Scientific), and 5 µL of extracted RNA. RT-qPCR reactions were run on a Rotorgene Q (QIAGEN) platform with the following thermal steps: 45° for 10 min, 95 °C for 10 min, followed by 45 cycles of 95 °C for 15 sec, 56 °C for 20 sec, and 72 °C for 30 sec [12]. Ct-values <42 were considered PPIV1 RNA positive.

## High-throughput qPCR

Extracted nucleic acids were initially reverse transcribed using a high-capacity cDNA RT Kit (Applied Biosystems). A final volume of 10 µL reaction mix was prepared by mixing 1 µL of 10X RT buffer, 0.4 µL dNTP mix (100 mM), 1 µL of 10X random hexamer, 0.5 µL of MultiScribe RT enzyme, 2.1 µL of nuclease-free water, and 5 µL of extracted nucleic acid. The cDNA synthesis was carried out in a PCRmax Alpha thermocycler (Cole-Parmer, Vernon Hills, IL, USA) with the following thermal conditions: 25 °C for 10 min, 37 °C for 120 min, and 85 °C for 5 min. Afterwards, cDNA samples were pre-amplified using 2X TaqMan PreAmp master mix (Applied Biosystems). A total volume of 10 µL was prepared by mixing 2.5 µL of cDNA with 5 µL of 2X TaqMan PreAmp master mix (Applied Biosystems), and 2.5 µL of primer mix (200 nM, containing all sets of primers).

The primers and probes used are published elsewhere and allowed the detection of: *Mycoplasma hyorhinis*, *Streptococcus suis type 2, Haemophilus parasuis (Glaesserella parasuis) Mycoplasma hyopneumoniae, Actinobacillus pleuropneumoniae, Bordetella bronchoseptica, Pasteurella multocida,* IAV (Inf M, Nagy2), porcine circovirus 2 and 3 (PCV2 and PCV3), porcine cytomegalovirus (PCMV), and porcine respiratory corona virus (PRCV) [19,27]. The pre-amplification was carried out in a PCRmax Alpha thermocycler (Cole-Parmer) using the following thermal cycling program: 95°C for 10 min

followed by 14 cycles of 95°C for 15 s and 60°C for 4 min. The pre-amplified products were stored at -20°C until further use. For high-throughput qPCR analysis, the BioMark HD (Standard BioTools, South San Francisco, CA, USA) and the 192.24 Dynamic array (DA) integrated fluidic circuit (IFC) chip (Standard BioTools) were used. A 4 µL sample mix was prepared for each of the samples by mixing 2.2 µL pre-sample mix (prepared by mixing 2 µL of 2X TaqMan Gene Expression Mastermix (Applied Biosystems) and 0.2 µL of 20X sample loading reagent (Standard BioTools) with 1.8 µL of the pre-amplified sample. Each PCR assay mix was prepared by mixing 2 µL primer/probe stock (containing 33 µM of each primer and 10 µM of probe) with 2 µL of 2X assay loading reagent (Standard BioTools). Three µL of assay mix and 3 µL of sample mix were loaded into the respective inlets of the 192.24 DA IFC chip.

The 192.24 DA IFC chip was placed in the IFC controller RX (Standard BioTools) for loading and mixing for approximately 30 min. Finally, the chip was inserted into the high-throughput qPCR platform BioMark HD (Standard BioTools) for thermal cycling with the following cycling conditions: 50°C for 2 min, 95°C for 10 min, followed by 40 cycles of 95°C for 15 s and 60°C for 60 s. In each chip run, positive and negative (nuclease-free water) controls were included. Amplification curves and Ct-values were obtained on the BioMark HD system and finally, analyzed using the high-throughput qPCR Analysis software 4.8.1 (Standard BioTools).

## Detection of PPIV1 NP RNA *in situ* by RNAscope

To detect PPIV1 RNA in FFPE tissues, *in situ* hybridization (ISH) was performed using the RNAscope 2.5 HD BROWN kit (Advanced Cell Diagnostics, Bio-techne, Ireland, Dublin) according to the manufacturer's instructions [28]. Briefly, a 9ZZ probe named V-PRV1-NP (catalog 1784821-C) targeting PPIV1 nucleoprotein (NP) RNA (antisense) was designed and synthesized by Advanced Cell Diagnostics. Tissue sections were deparaffinized with xylene, followed by two 99% ethanol washes and then hydrogen peroxidase blocking for 10 min. The slides were washed with Milli-Q water and pre-treated using kit-provided antigen retrieval buffer (boiled in water bath for 15 min) and protease plus (30 min). ISH signal was developed using the kit-provided pre-amplifier and amplifier conjugated to alkaline phosphatase and incubated with a DAB substrate solution for 10 min at room temperature. Sections were then counterstained with hematoxylin. This was performed on upper tracheal and lung tissues with PPIV1 RT-qPCR Ct-values <35, i.e., upper tracheal tissues from all PPIV1 pigs and three recipient pigs, lung tissues from two PPIV1 pigs, and the cranial tracheobronchial lymph node from a PPIV1 pig (two samples were missing from the recipient pigs).

## Sequencing

The German PPIV1 isolate used for inoculation was sequenced via the following protocol. After a centrifugation step of 10 min at 10,000×*g*, 140 uL of the supernatant was used for total nucleic acid isolation, performed with the Qiamp viral RNA mini kit (QIAGEN). A DNAse treatment was then performed on 11 µl of NA solution (DNAse I, New England Biolabs (NEB), Ipswich, MA, USA) (according to manufacturer's specifications) and the remaining RNA was subjected to reverse transcription (ProtoScript II First Strand cDNA Synthesis Kit, NEB) and second strand synthesis (NEBNext Ultra II Non-Directional RNA Second Strand Synthesis Module, NEB). Prepared dsDNA was finally purified with MAGBIO magnetic beads 1:1:v:v (HighPrep PCR-DX, MAGBIO, Gaithersburg, MD, USA) and outsourced to Novogene (Munich, Germany) for Illumina sequencing. Obtained reads were quality-controlled and trimmed using BBDuck in Geneious Prime (Dotmatics, Boston, MA, USA), and surviving reads were mapped to a reference PPIV1 sequence using Bowtie. Finally, the obtained contig was visually inspected, manually polished, and ambiguities were inserted where appropriate.

## Phylogenetic analysis and genomic comparison

A maximum-likelihood phylogenetic tree of the PPIV1 contained in the inoculum used in this study, together with 13 other PPIV1 sequences available in GenBank and including the virus in the inoculum used in Welch et al. 2021, was

constructed by IQ-TREE version 3.0 [5,29] using a 15,140 nts MAFFT-generated alignment. Mid-point rooting was conducted. The Modelfinder function in IQTREE 3.0 [30] was used to determine the best fitting substitution model [31–35]. Tree topology reliability evaluation was performed by bootstrap tests with 1000 pseudo-replicates [36]. The same alignment was used for direct nt and amino acid (aa) comparison of the sequences of the different isolates (the complete alignment is available in S2 File).

## Results

### Presence of other relevant respiratory pathogens at inoculation

At 0 DPI, nasal swab samples were investigated for the presence of other relevant respiratory pathogens by high-throughput qPCR. The analysis revealed that five pigs were positive for *M. hyorhinis* (one control, one PPIV1, one SOV, and two recipient pigs), two were positive for *S. suis* type 2 (two PPIV1 pigs), and one was positive for *G. parasuis* (PPIV1 pig). The specific pigs and Ct-values are listed in S2 Table. Nasal swab samples from all the pigs were negative for *M. hyopneumoniae*, *A. pleuropneumoniae*, *B. bronchoseptica, P. multocida,* IAV, PCV2, PCV3, PCMV, and PRCV.

### Unsuccessful SOV inoculation

The isolation of SOV was unsuccessful after two passages in both cell lines (LLC-MK2 and Calu-3 cells). No cytopathic effect was observed in the cells, and the cell suspension was negative for SOV RNA. Therefore, the final SOV inoculum consisted of 21 SOV-positive nasal swab samples, pooled and filtered, with a Ct-value of 20.72 in the RT-qPCR analysis. After inoculation no SOV RNA was detected in nasal swabs from any of the SOV-pigs at 1, 2, or 4 DPI, and therefore, four pigs were used as PPIV1 recipients to investigate horizontal transmission (Fig 1).

A high-throughput qPCR analysis of the SOV inoculum revealed the presence of IAV (Ct-value of 23), PCV3 (Ct-value of 26), PCMV (Ct-value of 11), *S. suis* type 2 (Ct-value of 20), *M. hyorhinis* (Ct-value of 19), *B. bronchiseptica* (Ct-value of 21) and *G. parasuis* (Ct-value of 13). Serous nasal discharge was observed in five pigs (546, 550, 576, 659, 666) at 2 DPI.

### Successful PPIV1 inoculation

One PPIV1 pig (512) showed serous nasal discharge at 4 DPI, and two PPIV1 pigs (512, 612) plus one recipient pig (659) showed mucopurulent nasal discharge at 11 DPI and 7 DPC, respectively. Otherwise, no clinical signs were observed during the study.

No PPIV1 or SOV nucleic acids were detected in nasal swabs by RT-qPCR nor in the tissues of the control group at any time point during the study (Fig 2A).

The PPIV1 inoculated pigs tested positive for PPIV1 RNA from 1 DPI until 11 DPI, with the lowest Ct-value (highest viral load) observed between 4 and 7 DPI (Fig 2A). The recipient pigs were moved into the pen at 0 DPC (4 DPI) and PPIV1 RNA shedding was detected at 3 DPC and continued until euthanasia at 10 DPC (Fig 2A). The highest viral shedding in the recipient pigs was observed at 7 DPC. In the PPIV1 pigs, the highest viral load was observed in tissues of the upper and lower trachea at 4 DPI, followed by the ventral meatus swab and BALF, as only two pigs were PPIV1 RNA positive in selected lung tissues (Fig 2C). At 10 DPC, all recipients were PPIV1 RNA negative in lung tissue samples, and samples from other tissues presented with Ct-values close to the limit of detection, except for one pig (507) in whose trachea and BALF samples PPIV1 was detected with Ct-values of ~20 (Fig 2D). At 14 DPI, PPIV1 RNA was detected in the lymph node and BALF of only one PPIV1 inoculated pig (658) (Fig 2E). Furthermore, one pig (505) tested positive for PPIV1 RNA in serum at 4 DPI with a Ct-value of 28.44.

PPIV1 was re-isolated from a sample collected from a recipient pig (659) at 7 DPC (nasal swab). Otherwise PPIV1 was not re-isolated in the selected samples including the PPIV1 RNA-positive serum sample. Overall results are summarized in Table 1.

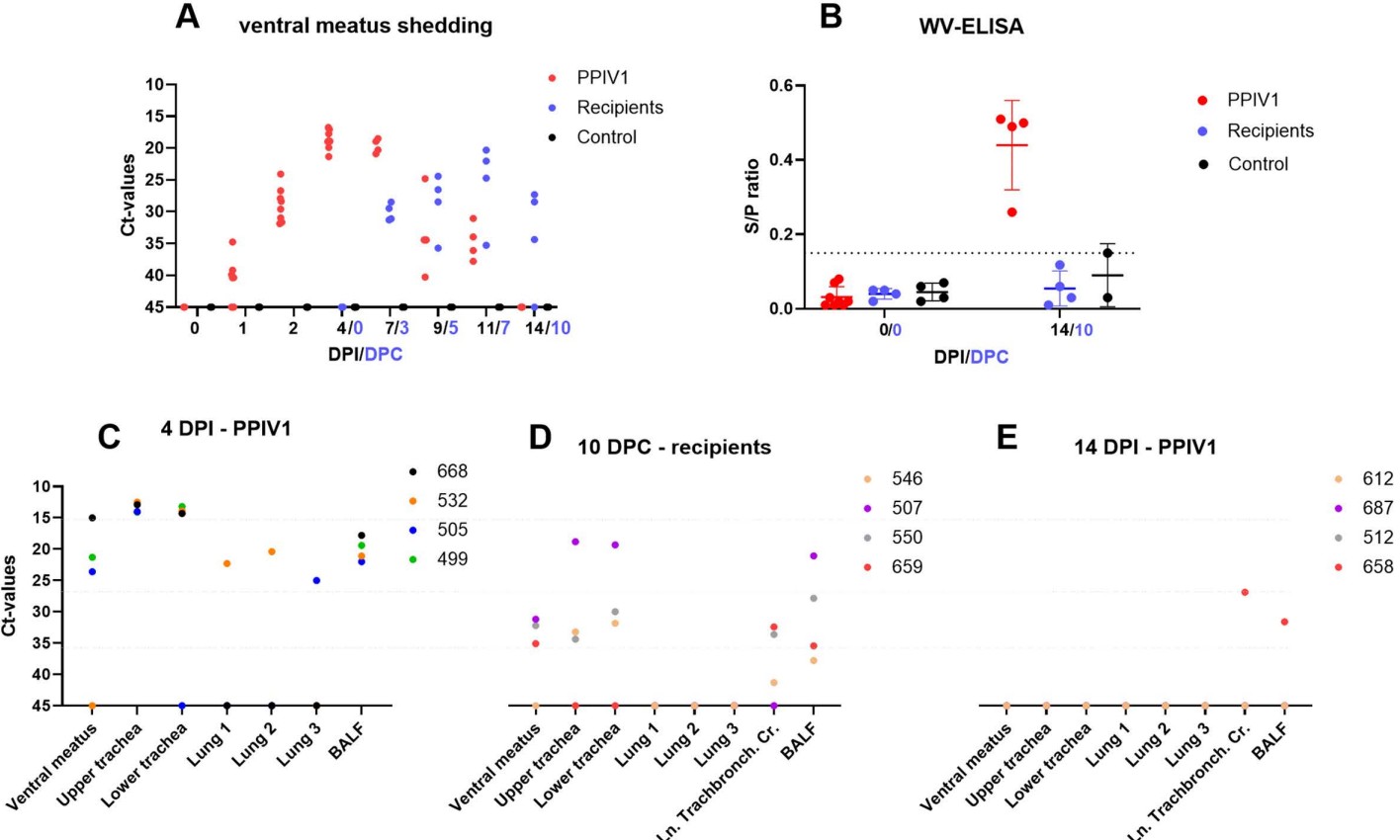

**Fig 2. Scatterplot of PPIV1 RT-qPCR nucleic acid detection (A, C, D, E) with Ct-values indicated on the y-axis and WV-ELISA (B) with sample/positive (S/P) ratio on the y-axis and mean with standard deviation presented.** In A) and **B)**, days post inoculation (DPI) or days post contact (DPC) are on the x-axis. In **C)**, d) and **E)**, the x-axis represents the tissue type at different time points. The dots indicate the value for each pig: A) ventral meatus nasal swab, C) tissue samples at 4 DPI from PPIV1 pigs, D) tissue samples at 10 DPC from recipients, and E) tissue samples at 14 DPI from PPIV1 pigs. Ln. Tracheobronch. Cr. = *Ln. tracheobronchialis cranialis*, BALF = bronchoalveolar lavage fluid. The graphs were made using GraphPad Prism.

## PPIV1 pigs seroconverted at 14 DPI

At 14 DPI, all the PPIV1 pigs ($n = 4$) seroconverted, and one control pig had a sample/positive (S/P) value at 0.15 which is 0.01 S/P above the cut-off, but PPIV1 nucleic acid was never detected in this pig (Fig 2B).

## None-to-mild macroscopic lung lesions

At 4 DPI, no macroscopic lesions were observed in the PPIV1 pigs, except for one pig (532) showing pulmonary rib impressions (Fig 3A). At 14 DPI, three pigs, one PPIV1 pig and two recipients (10 DPC), showed lesions consistent with chronic, lobular bronchopneumonia (658 (Fig 3B), 550 and 659), which affected 1.4 – 2.9% of the lung tissue. Pig 658 also showed mild hyperplasia of *In. tracheobronchialis cranialis*. Three pigs (two PPIV1 and one recipient) showed pulmonary rib impressions at 14 DPI/10 DPC (546, 612, 687). No macroscopic changes were observed in the SOV group.

## Microscopic changes observed in the trachea

During microscopic tissue evaluation at 4 DPI, 2/4 PPIV1 pigs (505 and 532) showed mild exudation of neutrophils, single cell necrosis, multifocal areas with loss of cilia, and infiltration of mononuclear cells in lamina propria of trachea (Fig 4).

**Table 1. Summary of results.**

| Group | Animal ID | Necropsy date | Nasal discharge | Peak PPIV1 RNA nasal shedding (Ct)/ PPIV1 isolation** | Highest PPIV1 RNA tissue load | PPIV1 RNA positive lung tissue/ lung lesions | PPIV1 serological status |
|---|---|---|---|---|---|---|---|
| Control | 616 | 4 DPI | No | NA | NA | NA | NA |
| | 569 | 4 DPI | No | NA | NA | NA | NA |
| | 670 | 14 DPI | No | NA | NA | NA | Positive[b] |
| | 569 | 14 DPI | No | NA | NA | NA | Negative |
| PPIV1 | 688 | 4 DPI | No | 17.8/ NA | Upper trachea | No/ No | NA |
| | 532 | 4 DPI | No | 21.3/ NA | Upper trachea | Yes/ Yes | NA |
| | 505 | 4 DPI | No | 19.9/ No[a] | Upper trachea | Yes/ No | NA |
| | 499 | 4 DPI | No | 18.9/ NA | Lower trachea | No/ No | NA |
| | 612 | 14 DPI | Yes | 17.1/ No | NA | No/ Yes | Positive |
| | 687 | 14 DPI | No | 19.0/ No | NA | No/ Yes | Positive |
| | 512 | 14 DPI | Yes | 19.0/ No | NA | No/ No | Positive |
| | 658 | 14 DPI | No | 16.8/ No | Ln. tb. cr. | No/ Yes | Positive |
| Recipient* | 546 | 14 DPI | Yes | 22.0/ No | Lower trachea | No/ Yes | Negative |
| | 507 | 14 DPI | No | 28.5/ No | Upper trachea | No/ No | Negative |
| | 550 | 14 DPI | Yes | 24.7/ No | Lower trachea | No/ Yes | Negative |
| | 659 | 14 DPI | Yes | 20.3/ Yes | Ln. tb. cr. | No/ Yes | Negative |
| SOV | 520 | 4 DPI | No | NA | NA | NA | NA |
| | 686 | 4 DPI | Yes | NA | NA | NA | NA |
| | 576 | 4 DPI | Yes | NA | NA | NA | NA |
| | 666 | 4 DPI | No | NA | NA | NA | NA |

*14 days post inoculation (DPI) corresponds to 10 days post contact.

** Performed on nasal swab samples with highest cycle threshold (Ct)-values. [a] Performed on a serum sample from 4 DPI.

[b] Close to cut-off and not PPIV1-positive in any samples from the study

NA= not applicable. Ln. tb. cr.= lymph nodulus tracheobronchialis cranialis.

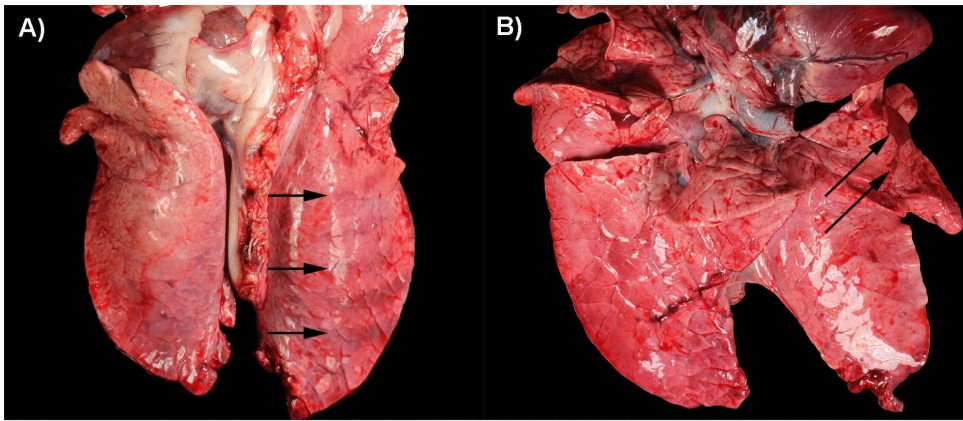

**Fig 3. Macroscopic changes observed in PPIV1 infected pigs. A)** Arrows indicate pulmonary rib impressions observed in a PPIV1 pig at 4 DPI (532). **B)** Chronic, lobular bronchopneumonia (arrows) observed in a PPIV1 pig at 14 DPI (658).

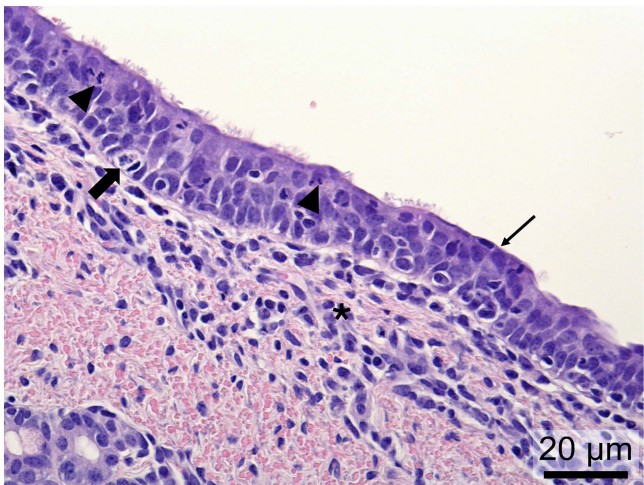

**Fig 4. Chronic tracheitis from a PPIV1-inoculated pig (505) at 4 days post inoculation (DPI).** Histopathological changes consisted of exudation of neutrophils (arrowheads), single cell necrosis (thick arrow), loss of cilia (thin arrow) and infiltration of mononuclear cells in lamina propria (asterisks).

One PPIV1 pig (499) showed similar lesions but with exudation consisting of mixed inflammatory cells and no infiltration of mononuclear cells in the lamina propria, also comparable to chronic tracheitis, and one pig (688) had a focal infiltration of mononuclear cells with multifocal areas of cilia loss.

At 14 DPI, all PPIV1 pigs and two recipient pigs (10 DPC) showed varying degrees of exudation with mixed inflammatory cells, epithelial erosion, loose tight junctions, and infiltration of mononuclear cells in the lamina propria (chronic, erosive tracheitis) (Fig 5). One of the remaining recipient pigs (546) showed chronic tracheitis (as defined for the PPIV1 pigs at 4 DPI), and the other (659) showed severe erosive changes (only the basal epithelial layer remaining) with suppurative exudation (S1 Fig).

Varying degrees of acute (no infiltration of mononuclear cells and exudation of neutrophils) to chronic tracheitis (like the PPIV1 pigs 4 DPI) were observed in the tracheal tissues of the SOV pigs at 14 DPI, but otherwise no lesions were observed.

No lesions were observed in the controls (S2 Fig), except for one pig (616) that showed multifocal areas in the tracheal epithelium with pustules.

In the lungs, two PPIV1 pigs at 14 DPI (pigs 512, 658) and one recipient pig at 10 DPC (pig 550) showed chronic bronchointerstitial pneumonia in one or two of the cranial lung samples at 14 DPI (Fig 6), otherwise, no lesions were observed except for one control pig (616) that showed interstitial pneumonia at 4 DPI.

### *In situ* detection of PPIV1 NP RNA in trachea and lung tissues

All PPIV1 pigs showed a high number of PPIV1-positive cells in the epithelial cells of the upper tracheal tissues at 4 DPI (Fig 7A). One PPIV1 pig (532) also showed a high amount of PPIV1-positive bronchiolar epithelial cells with a few positive cells in the alveoli (Fig 7B), whereas another PPIV1 pig (505) only showed a few positive cells in the alveoli. One recipient pig (507) showed PPIV1-positive cells in the epithelial cells of the upper tracheal tissues at 10 DPC.

### Genomic distance to the previously evaluated clade II PPIV1

Phylogenetic analysis revealed that the 14 included PPIV1 sequences separate into the two pre-defined PPIV1 clades, I and II (Fig 8). The German PPIV1 GER/2022AI03675/2022(SK398) (GenBank: PV767405) isolate evaluated in this

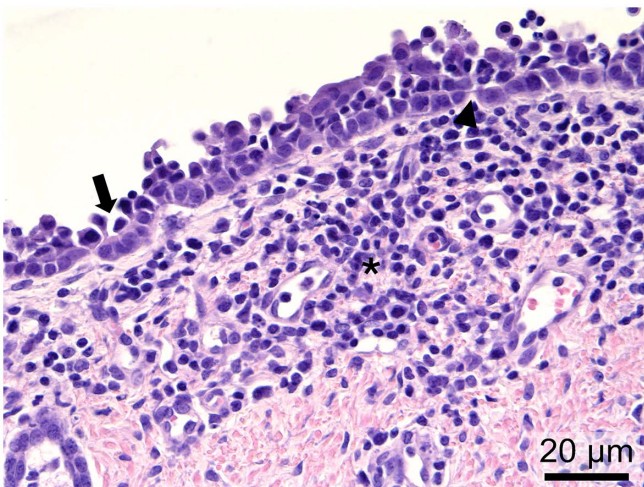

**Fig 5. Chronic, erosive tracheitis from a PPIV1-inoculated pig (658) at 14 days post infection (DPI).** Histopathological changes consisted of loose tight junctions (arrowhead), erosion of the epithelial layer (arrow) and infiltration of mononuclear cells in the lamina propria (asterisks).

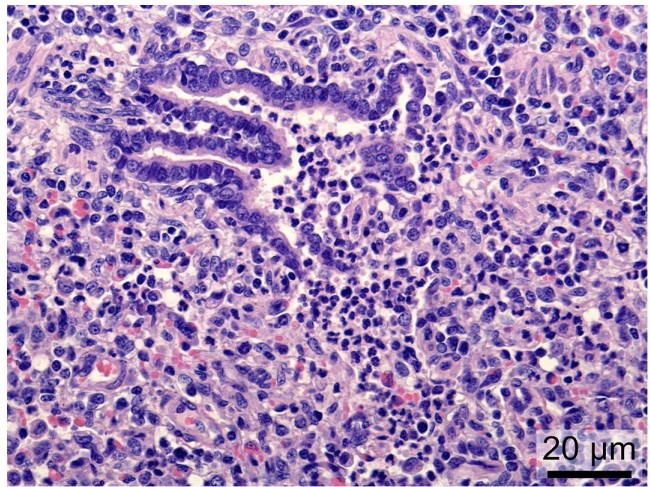

**Fig 6. Microscopic changes in the lungs of PPIV1-infected pigs.** Lungs from a PPIV1 pig (658) at 14 DPI showing exudation of neutrophils in a terminal bronchiole that exceeds into the lumen of the alveoli and infiltration of mononuclear cells in the interstitium. These findings are consistent with chronic, bronchointerstitial pneumonia.

study appeared in clade I, whereas the PPIV1 isolate evaluated experimentally by studies from Welch et al. (2021, 2022a, 2023) appeared in clade II. Evaluation of nucleotide (nt) and amino acid (aa) similarity between the two isolates used for two experimental PPIV1 inoculations (DE, clade I and US, clade II) disclosed that the highest difference appeared in the phosphoprotein (pairwise identity of 88.9% and 84.5% at nt and aa level, respectively) and partly also the surface proteins (90.0% and 92.4% nt and aa pairwise identity, respectively in the hemagglutinin/neuraminidase region, and 90.4% and 91.4% nt and aa identity in the fusion protein region, respectively; S3 Table). The nucleoprotein, matrix protein, and polymerase of the German isolate were all 92.0-92.6% and 96.0-96.1% identical to the PRV1/US/2016 isolate at nt and aa levels, respectively.

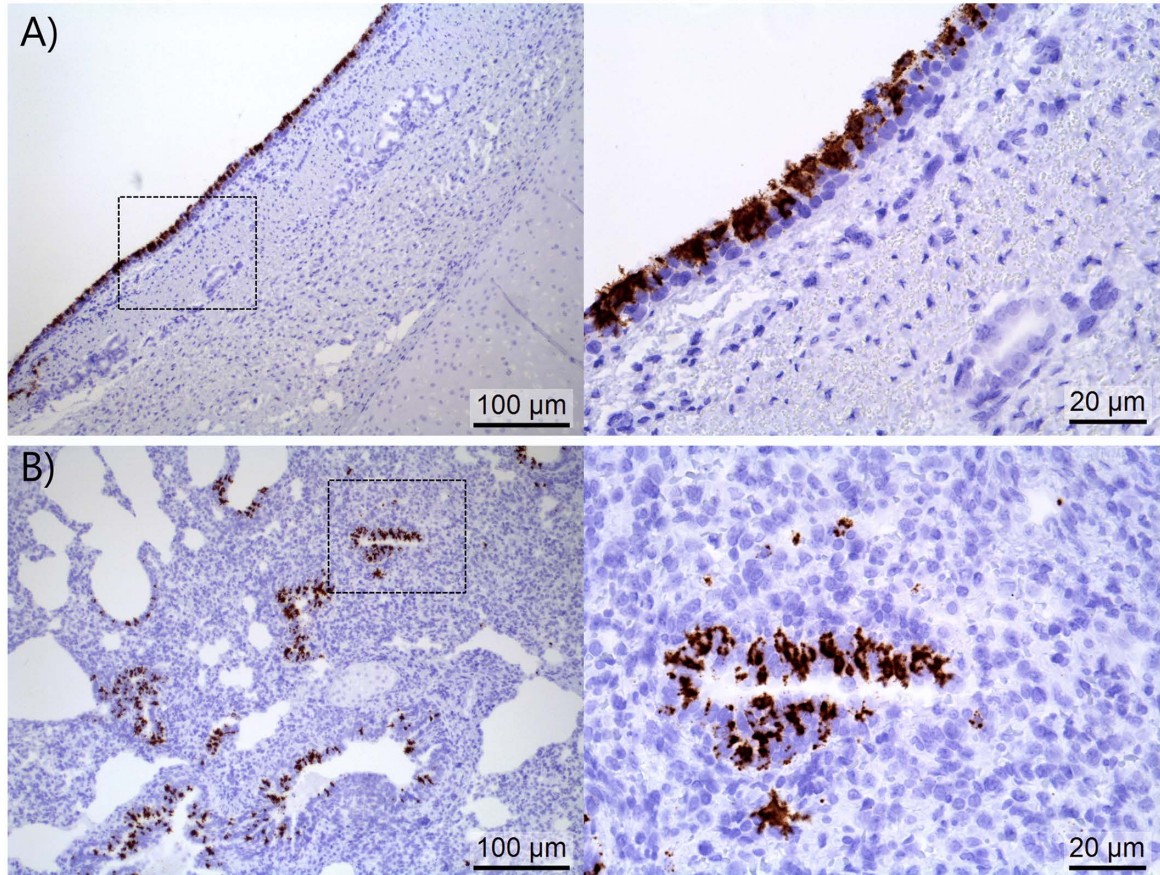

**Fig 7. PPIV1 positive epithelial cells in the upper trachea, bronchioles and alveoli. A)** A high number of PRV1-positive epithelial cells (brown) of the upper trachea in a PPIV1 pig at 4 DPI (499). **B)** A high amount of PPIV1-positive bronchiolar epithelial cells (close-up showing a PPIV1-positive terminal bronchiole) and a few in the alveoli of a PPIV1 pig at 4 DPI (532).

## Discussion

In this experimental trial, we demonstrated that five-week-old, PPIV1-seronegative pigs directly exposed to PPIV1 got infected and excreted PPIV1 RNA from the day after exposure and for 11 days. The inoculated pigs successfully transmitted the virus through direct contact to recipient pigs, resulting in sustained shedding of infectious virus. Although none of the exposed pigs exhibited respiratory clinical signs (other than nasal discharge), histopathology examination revealed chronic tracheitis at 4 DPI, which progressed to chronic, erosive tracheitis by 14 DPI. These findings appeared both in intranasally-inoculated pigs and in direct-contact pigs (horizontally infected pigs).

The role of PPIV1 in causing histopathological lesions was supported by *in situ* viral RNA detection, showing a strong signal in the upper tracheal epithelial cells of all pigs. PPIV1-positive cells were not completely restricted to epithelial cells in the URT, but the progression to bronchiolar epithelial cells and pneumocytes was limited. Observed histopathological changes (tracheitis and chronic bronchointerstitial pneumonia) did not lead to any observed macroscopic changes apart from pulmonary rib impressions or mild lobular bronchopneumonia in seven pigs, five of which showed nasal discharge as their only clinical manifestation. Overall, these results, combined with the findings of Welch et al. [5,7,8], prove that PPIV1 is a primary porcine respiratory pathogen.

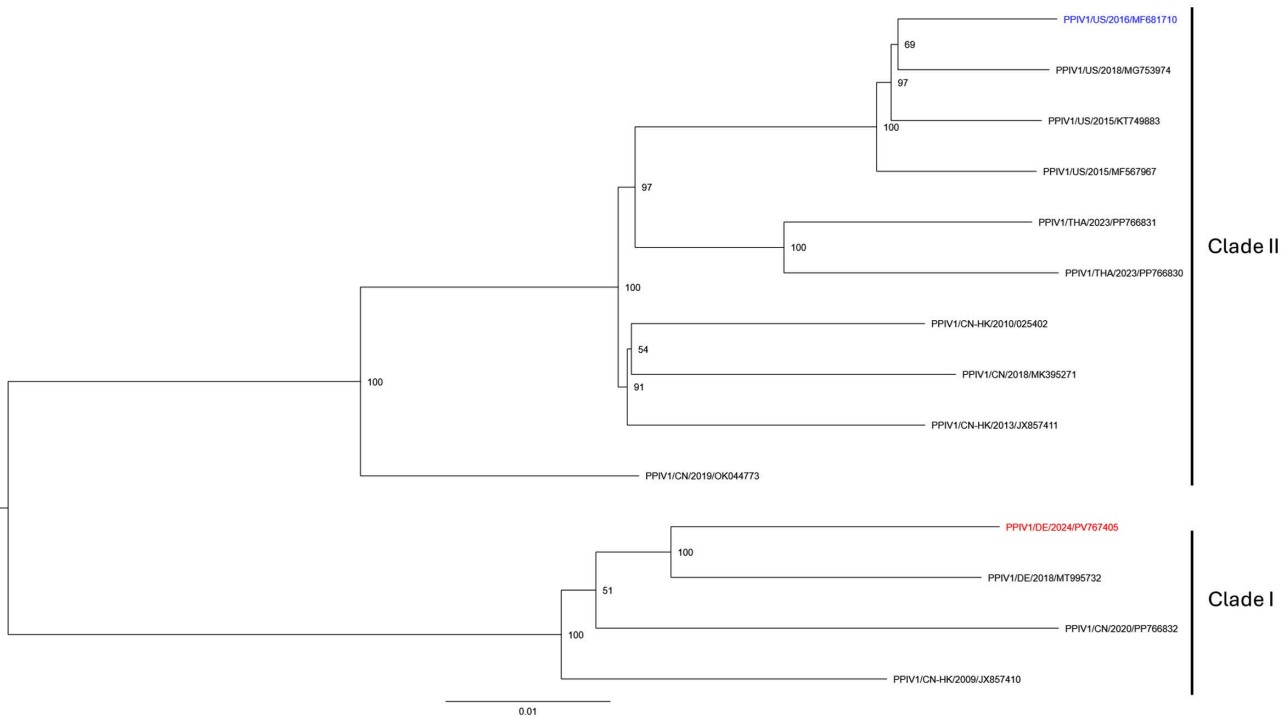

**Fig 8. A phylogenetic tree of 14 PPIV1 genomes (nt 120-12,260).** The inoculum from this study (red) and an experimental inoculum from Welch et al., 2021 (blue) are included. The maximum-likelihood phylogenetic tree was built with IQ-TREE 3.0 according to the general time reversible model with empirical base frequencies and four discrete rate category gamma-distributed rate heterogenity (GTR + F + G4). The outcome of the bootstrap analysis (1000 replicates) used to test branch robustness, is shown for each branch.

PPIV1 is frequently detected in pigs with respiratory signs, but often in combination with other pathogens [12,13]. PPIV1 has an extensive effect on the epithelial lining, which is regarded as a significant part of the innate immune defense [37,38], thereby compromising significantly this mechanical barrier against pathogens. It has also been proven how the breakage of epithelial integrity provide opportunity for further infection with *A. pleuropneumoniae* [39].

The degeneration of the epithelial integrity by PPIV1 may therefore be highly considered to facilitate secondary bacterial infections, thus contributing to the porcine respiratory disease complex (PRDC) [3]. This is also indirectly proven by the conventional group of pigs included in the study of Welch et al. 2021, because their PPIV1-associated lesions in the presence of co-infecting bacteria were different compared to pigs without bacterial detections. Interestingly, experimental co-infection with another virus, IAV, did not worsen PPIV1 symptoms [8]. On the contrary, the authors found that co-infections with IAV reduced PPIV1 nucleic acid load in the URT. The two viruses also showed different tissue tropisms in the respiratory tract of pigs. PPIV1 primarily showed tropism for the URT, whereas IAV also showed tropism for the lower respiratory tract (LRT, bronchi, bronchioles, and alveoli) with multiple studies showing IAV-induced pneumonia [40–42]. The host receptor of PPIV1 remains unknown, but human respiroviruses bind to sialic acids (SA) linked to *N*-Acetyllactosamine (Galβ1–4GlcNAc) by either α2,3 or α2,6 linkages (SA-α2,3/6-Galβ1–4GlcNAc), and swine IAVs mainly bind to SA-α2,6-Gal [43,44]. Therefore, a potential competition of host receptor between PPIV1 and IAV could explain the reduced PPIV1 antigen detection in co-infected pigs from Welch et al., 2023.

The findings observed in this study are comparable to the findings of the CDCD group of pigs in Welch et al. 2021 (US, clade II) [5] and those described for other viruses in the same family as PPIV1, such as bovine parainfluenza and human parainfluenza viruses, which cause pharyngitis, croup (laryngotracheobronchitis), and pneumonia [5,45–47].

However, in our study, PPIV1 was sparsely detected in the LRT, and only a few pigs developed pneumonia. This difference in involvement of the LRT could be explained by the CDCD origin of the pigs in the study of Welch et al. 2021 [5], contrary to conventional Norwegian SPF pigs with none-to-few influential secondary pathogens (*M. hyorhinis, S. suis type 2, G. parasuis*), recognized by high-throughput qPCR, in our study. Genomic difference (9,1% nt difference) between the isolates used in the two studies could also be relevant. Indeed, the PPIV1 isolate used in this study is from clade I, which is different from the clade II PPIV1 isolate used in the studies of Welch et al., [5,7,8]. The aa variation observed between clades I and II isolates could impact the virulence of these viruses since the primary variation appears in the ORFs for the phosphoprotein and the surface protein (hemagglutinin/neuraminidase and the fusion genes). Nonetheless, the knowledge about this virus is still limited, and additional sequencing efforts and pathogenicity studies will be required to identify molecular determinants of disease severity.

Serous nasal discharge was reported in 5/8 pigs at 2 DPI, and lesions were observed in the SOV pigs at 4 DPI, whereas no clinical signs or lesions were observed in the controls at 4 and 14 DPI (except one control pig showing distinct lesions at 4 DPI), indicating a potential effect of other pathogens present in the SOV inoculum. This could also explain why the two PPIV1 recipient pigs (which received the SOV inoculum at 0 DPI) showed different lesions compared to the PPIV1 pigs. However, it is worth mentioning that, due to the pre-amplification step, the Ct-values reported for the high-throughput qPCR are, in general, lower (around 8–12 Ct-values) compared to Ct-values obtained by traditional RT-qPCR platforms. The presence of other respiratory pathogens was also detected in BALF samples from conventional pigs in the study from Welch et al. 2021 [5]. Therefore, some of the lesions observed in this study might be due to the presence of opportunistic pathogens or the synergistic effect of multiple pathogen infections.

The serum of one control pig was mildly reactive in PPIV1 wv-ELISA (0.149), but we argue that the sample should be regarded as an ELISA false positive. Although low-level cross-reactivity with antibodies against conserved paramyxovirus antigens cannot be completely excluded when using a whole-virus ELISA, the high assay specificity at the selected cutoff (0.15), and the absence of virological evidence indicate that this result most likely reflects non-specific ELISA reactivity rather than true PPIV1 seroconversion.

A productive PPIV1 infection in the inoculated pigs was confirmed by the rapid (1–2 days) horizontal infection to recipient pigs, confirmed by PPIV1 nasal shedding and seroconversion at 14 DPI. The pigs had direct contact, and therefore, it is not possible to determine the exact route of transmission. PPIV1 was re-isolated from one nasal swab of a recipient pig collected at 7 DPC, confirming a successful transmission. PPIV1 was not re-isolated from the rest of the samples, which might be due to sample handling (freeze/thaw cycles), as these samples were not intended for culturing. The finding of PPIV1 RNA in the serum of only one pig and the lack of re-isolation indicate that PPIV1 causes a local infection in the respiratory tract with likely oronasal transmission. At 10 DPC, it was no longer possible to isolate viable viruses from nasal swabs, and only high Ct values were detected by RT-qPCR. This suggests that the virus is transmissible for approximately up to nine days after exposure, with a possible peak at 4 DPI. This conclusion is also supported by the overall RT-qPCR detections in respiratory tissues. Similar to handling of the samples for PPIV1 re-isolation, the PPIV1 inoculation isolate was also frozen/thawed before performing the RT-qPCR-based TCID50/ml, potentially resulting in an underestimation of the inoculation titer.

More knowledge about the immunological response against PPIV1, as well as the implications of co-infections, should be investigated. Understanding these interactions may clarify factors leading to systemic infections and/or more severe tissue damage with greater clinical impact on the piglet health. Until future vaccines against PPIV1 become available, or until autogenous vaccines are provided, management practices limiting respiratory pathogens spread should be implemented to reduce PPIV1 impact on porcine health. These include sectioning, increased air exchange, and appropriate stocking density.

The attempt to propagate SOV in LLC-MK2 or Calu-3 cells was unsuccessful, as was the case in another study by Graaf-Rau et al. [12] as well as in an attempt to isolate a novel feline orthopneumovirus in A-72 (canine tumor fibroblast)

cells [48], indicating that not all orthopneumoviruses can be cultured in commonly available cell lines. The competence of other cell lines for SOV should be investigated since using cell culture supernatant, instead of pooled SOV positive nasal swabs, could significanlty improve the success of experimental SOV infections and reduce the chances of acquiring additional infections. Isolation of SOV using Hep-2, A549, Vero, and HeLa cell lines should be explored since these have previously been used to isolate HRSV [49–51]. The lack of successful SOV inoculation of pigs could be due to the final SOV inoculum (pooled nasal swabs) not containing sufficient viable replicative virus for successful infection, the pigs having antibodies capable of neutralizing SOV, or the virus being less efficient at establishing infection in pigs. Further research on SOV is needed, including investigations to determine the clinical relevance and distribution of this virus in pigs.

## Conclusion

PPIV1 was confirmed to be a primary porcine respiratory pathogen that caused no clinical signs, except for nasal discharge, in experimental monoinfections. PPIV1 primarily affected epithelial cells in the URT, resulting in chronic, erosive tracheitis. PPIV1 RNA was detected from 1 to 11 DPI and was transmitted horizontally to direct-contact pigs. Given the pathological manifestations, PPIV1 should be considered a pathogen that predisposes to other potential infections. In contrast, SOV could not be isolated in cells or experimentally inoculated into pigs for reasons that are yet to be elucidated.

## Supporting information

**S1 File. Raw data presented in Fig 2. Negative Ct-values are reported as 45.**
(ZIP)

**S2 File. Complete sequence alignment of the PPIV phylogenetic analysis.**
(ZIP)

**S1 Fig. Trachea from a recipient pig at 14 days post inoculation (DPI) chronic, severe, erosive tracheitis.** The histopathological changes observed were suppurative exudation, with disseminated epithelial erosion leaving only the basal cell layer and with infiltration of mononuclear cells in lamina propria.
(TIF)

**S2 Fig. Trachea from a control pig at 4 days post inoculation (DPI) with no histopathological changes observed.**
(TIF)

**S1 Table. Overview of the pigs that were used in the experiment including group name, ID, gender, date of euthanization, weight (W) at -7 days post inoculation (DPI), 0, 4 and 14 DPI.**
(XLSX)

**S2 Table. Presence of other relevant respiratory microbes in nasal swabs collected at 0 DPI investigated by high-throughput qPCR.**
(XLSX)

**S3 Table. Pairwise sequence identities comparing the 15.320 nt long genome and single ORFs of PPIV1-isolates used in this study (clade I) and in studies from Welch et al., 2021, 2022a, 2023 (clade II).**
(XLSX)

## Acknowledgments

We want to thank Nina Dam Grønnegaard, Hue Thi Thanh Tran, Elisabeth Wairimu Petersen, and Betina Gjedsted Andersen for practical laboratory help and to Dennis Bork for figure editing. Furthermore, we wish to thank Bjørg Skovmand Helstad, Maja Rosendal, and Karen Martiny for their help with the animal experiments. Lastly, we wish to thank Amanda

Øpstun Birk, Christoffer Kirkelund Flyger, Frederik Andersen, Denis Koylyu, Sophie Joanna George for practical help with the necropsies.

## Author contributions

**Conceptualization:** Marianne Viuf Agerlin, Kasper Pedersen, Lars Erik Larsen, Charlotte Kristensen.

**Formal analysis:** Marianne Viuf Agerlin, Kasper Pedersen, Charlotte Kristensen.

**Funding acquisition:** Henrik Elvang Jensen, Lars Erik Larsen.

**Investigation:** Marianne Viuf Agerlin, Kasper Pedersen, Mathias Romar, Marta Canuti, Charlotte Kristensen.

**Methodology:** Marianne Viuf Agerlin, Kasper Pedersen, Lars Erik Larsen, Charlotte Kristensen.

**Project administration:** Marianne Viuf Agerlin, Kasper Pedersen, Charlotte Kristensen.

**Resources:** Mathias Romar, Timm Harder, Nicole Bakkegård Goecke.

**Supervision:** Marta Canuti, Nicole Bakkegård Goecke, Henrik Elvang Jensen, Lars Erik Larsen, Pia Ryt-Hansen, Charlotte Kristensen.

**Visualization:** Marianne Viuf Agerlin, Kasper Pedersen, Charlotte Kristensen.

**Writing – original draft:** Marianne Viuf Agerlin, Kasper Pedersen, Charlotte Kristensen.

**Writing – review & editing:** Mathias Romar, Marta Canuti, Timm Harder, Nicole Bakkegård Goecke, Henrik Elvang Jensen, Lars Erik Larsen, Pia Ryt-Hansen.

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
