## [Decision Letter · Decision Letter 0]

18 Nov 2025

PPATHOGENS-D-25-01809

Experimental inoculation of pigs with porcine respirovirus type 1 revealed pathological manifestations in the upper respiratory tract

PLOS Pathogens

Dear Dr. Kristensen,

Thank you for submitting your manuscript to PLOS Pathogens. After careful consideration, we feel that it has merit but does not fully meet PLOS Pathogens's publication criteria as it currently stands. Therefore, we invite you to submit a revised version of the manuscript that addresses the points raised during the review process.

We look forward to receiving your revised manuscript.

Kind regards,

Amy L. Hartman, PhD

Academic Editor

PLOS Pathogens

Kanta Subbarao

Section Editor

PLOS Pathogens

Sumita Bhaduri-McIntosh

Editor-in-Chief

PLOS Pathogens

orcid.org/0000-0003-2946-9497

Michael Malim

Editor-in-Chief

PLOS Pathogens

orcid.org/0000-0002-7699-2064

**Journal Requirements:**

1) We do not publish any copyright or trademark symbols that usually accompany proprietary names, eg ©,  ®, or TM  (e.g. next to drug or reagent names). Therefore please remove all instances of trademark/copyright symbols throughout the text, including:

- ® on page: 13.

3) Some material included in your submission may be copyrighted. According to PLOSu2019s copyright policy, authors who use figures or other material (e.g., graphics, clipart, maps) from another author or copyright holder must demonstrate or obtain permission to publish this material under the Creative Commons Attribution 4.0 International (CC BY 4.0) License used by PLOS journals. Please closely review the details of PLOSu2019s copyright requirements here: PLOS Licenses and Copyright. If you need to request permissions from a copyright holder, you may use PLOS's Copyright Content Permission form.

Potential Copyright Issues:

i) Please confirm (a) that you are the photographer of Figure 3., or (b) provide written permission from the photographer to publish the photo(s) under our CC BY 4.0 license.

4) Please amend your detailed Financial Disclosure statement. This is published with the article. It must therefore be completed in full sentences and contain the exact wording you wish to be published.

**Reviewers' Comments:**

Reviewer's Responses to Questions

**Part I - Summary**

Reviewer #1: I appreciated the opportunity to review this manuscript on the experimental inoculation of pigs with porcine respirovirus type 1 (PRV1) and swine orthopneumovirus (SOV). In this study, the authors examined viral shedding and pathology associated with these infections. The study is well designed and clear, contributing to our understanding of porcine pathogens with potential health and economic consequences for the livestock industry.

Reviewer #2: The article deals with an important study of experimental infection showed that PRV1 induces nasal discharge, primarily affects the upper respiratory tract, causes tracheitis, and transmits efficiently by direct contact. In contrast, SOV inoculation did not result in detectable infection. Overall, the paper might be interesting for researchers working with PRV and SOV. However, before acceptance, the manuscript requires editing with the minor points.

Reviewer #3: Agerlin, Pedersen, et al. present an interesting, informative, and well-designed study to assess the pathology of novel porcine respiratory viruses PRV1 and SOV. These newly identified porcine respiratory pathogens are of great clinical importance and this inoculation and transmission study is an important addition to the field. The introduction and discussion are well written and concise. My main criticism is that the results section is a bit sparse and at times hard to follow and a lot of information is cached in the materials and methods section. The manuscript could benefit from some minor editing and reformatting to move a bit more information from the materials and methods into the results section so that it is easier to follow for the reader. Otherwise this paper was interesting and enjoyable to read and I look forward to learning more about the pathogenesis of these viruses in the future!

**Part II – Major Issues: Key Experiments Required for Acceptance**

Please use this section to detail the key new experiments or modifications of existing experiments that should be absolutely required to validate study conclusions.required to validate study conclusions.required to validate study conclusions.required to validate study conclusions.

Reviewer #1: The authors mention on lines 346-348 and 467-469 that a pig in the control group had a positive ELISA result without detection of PRV1 RNA, and they suggest that this represents a false positive. How specific is this serological assay and could a positive result be the product of cross-reactivity (suggesting an infection with a different, related antigen)? The authors state that the pigs were negative by ELISA on arrival, but is it possible that seroconversion for a different infection was delayed and appeared during the study?

I suggest that the authors submit their raw alignments (lines 301-302) to ensure the repeatability of their genomic analyses.

I strongly suggest that the authors include a table summarizing the results and timeline of the PCR, serological, and isolation detection attempts for all pigs enrolled in the study; this would make the results easier to follow.

Reviewer #2: (No Response)

Reviewer #3: 1. Some of the information in the study design section of materials and methods would be more helpful in results section for readability. Figure 1 is also not referenced in the results section which I believe is a requirement of the journal. Moving some additional information into results would help with the requirement to reference figure 1 in the text.

a. Line 313 could use some detail/clarification and additional information on attempts to rescue SOV and infect animals would be helpful.

**Part III – Minor Issues: Editorial and Data Presentation Modifications**

Reviewer #1: Lines 59-66: In the introduction, consider including some background context on Nipah virus, another paramyxovirus that was identified after a large outbreak on a pig farm with devastating economic consequences for the industry (and onward zoonotic transmission to humans). This will help motivate the focus of this study.

Relatedly, has the zoonotic potential of PRV1/SOV been studied? Would be worth mentioning whether this has been assessed.

I would also consider mentioning the possibility of coinfection and PRV1 infection facilitating other pathogens; this is explored in the discussion but would help motivate the focus of the study (and testing for multiple pathogens) if included in the introduction.

Lines 77-78: more context about phylogenetic variation (or lack thereof) among PRV1 strains would be helpful here.

Line 118: where was the field study from which these SOV positive samples were collected?

Line 139 and table S1: I would use the word sex instead of gender.

In figure 1, why are the SOV-inoculated pigs two different colours and the PRV1-inoculated pigs one? Is this to indicate that half became recipients?

The last two paragraphs of the discussion as well as the conclusion should be proofread for grammar and spelling.

The SOV results are discussed only briefly in the discussion and not at all in the conclusion. While inoculation was unsuccessful, I think this is a point worth highlighting and discussing the implications of, as it could indicate that this virus is less effective at producing infections in pigs.

Reviewer #2: Title and abstract

- Line 19 According to the ICTV, Porcine respirovirus type 1 (PRV1) has been renamed Respirovirus suis. The authors are kindly requested to update the title and the entire text accordingly, and to include the appropriate ICTV reference and release version.

Introduction

- Lines 87-90 In reference no. 5 (ICTV website), the date of consultation is missing. Moreover, in the Orthopneumovirus genus, the muris species should be listed together with HRSV and BRSV, as indicated in the current ICTV taxonomy. Please update the Introduction accordingly.

Materials and methods

- Line 112 Could the authors please clarify whether the inoculum was removed after the one-hour incubation period?

- Lines 115-116 The authors are kindly requested to provide the reference for the RT real-time PCR method employed to verify the presence of viral nucleic acid. Please specify whether PRV1 produced a cytopathic effect in the LLC-MK2 cells.

- Lines 117-124 Please clarify whether both cell lines, LLC-MK2 and Calu-3, were used in the attempt to isolate SOV, as this point is not clearly stated in the manuscript. In addition, please specify the cycle threshold (Ct) values observed as positive in the RT-qPCR, and indicate which RT-qPCR assay was used by providing the appropriate reference. Since these methods are mentioned here for the first time, the corresponding references should be provided.

- Line 136 Please provide the reference for the ELISA test used to detect anti-PRV1 antibodies.

- Line 142 The authors are kindly requested to specify which active substance was used for the anesthesia of the pigs.

- Figure 1 The depiction of the nasal swab at day 3 post-infection is missing and should be included.

- We suggest inserting Table S1 directly into the main text, as this would make the identification of the animals used in the experiment much clearer.

Reviewer #3: 1. You compare European and North America PRV1 strains in line 70. It was unclear to me as a reader that we did not know how divergent these strains are (this is explored in Fig 8. Rewriting this to highlight that you will investigate this would be helpful.

2. In paragraph line 85 you note that SOV was discovered recently, is there a year for this?

3. Line 87 “SOV, which did not yet receive an official species designation” - should read “which has no yet received an official species designation”

4. Line 497- should be “chances” instead of “changes"

PLOS authors have the option to publish the peer review history of their article (what does this mean?). If published, this will include your full peer review and any attached files.). If published, this will include your full peer review and any attached files.). If published, this will include your full peer review and any attached files.). If published, this will include your full peer review and any attached files.

...

Reviewer #1: No

Reviewer #2: No

Reviewer #3: No

**Figure resubmission:**
---

## [Editor Report · Decision Letter 1]

29 Jan 2026

PPATHOGENS-D-25-01809R1

Experimental inoculation of pigs with porcine parainfluenza virus 1 revealed pathological manifestations in the upper respiratory tract

PLOS Pathogens

Dear Dr. Kristensen,

Thank you for submitting your manuscript to PLOS Pathogens. After careful consideration, we feel that it has merit but does not fully meet PLOS Pathogens's publication criteria as it currently stands. Therefore, we invite you to submit a revised version of the manuscript that addresses the points raised during the review process.

We look forward to receiving your revised manuscript.

Kind regards,

Amy L. Hartman, PhD

Academic Editor

PLOS Pathogens

Kanta Subbarao

Section Editor

PLOS Pathogens

Sumita Bhaduri-McIntosh

Editor-in-Chief

PLOS Pathogens

orcid.org/0000-0003-2946-9497

Michael Malim

Editor-in-Chief

PLOS Pathogens

orcid.org/0000-0002-7699-2064

**Additional Editor Comments:**

The abstract should be edited significantly to focus on the high level results and take home message of the study. As written, the abstract has too many experimental details (such as the n per group) that are best left for the main manuscript. The author summary is reasonable as written.

**Journal Requirements:**

**Reviewers' Comments:**

**Figure resubmission:**
---

## [Editor Report · Decision Letter 2]

5 Mar 2026

Dear Ms Kristensen,

We are pleased to inform you that your manuscript 'Experimental inoculation of pigs with porcine parainfluenza virus 1 revealed pathological manifestations in the upper respiratory tract' has been provisionally accepted for publication in PLOS Pathogens.

Best regards,

Amy L. Hartman, PhD

Academic Editor

PLOS Pathogens

Kanta Subbarao

Section Editor

PLOS Pathogens

Sumita Bhaduri-McIntosh

Editor-in-Chief

PLOS Pathogens

orcid.org/0000-0003-2946-9497

Michael Malim

Editor-in-Chief

PLOS Pathogens

orcid.org/0000-0002-7699-2064
---

## [Editor Report · Acceptance letter]

Dear Ms Kristensen,

We are delighted to inform you that your manuscript, "Experimental inoculation of pigs with porcine parainfluenza virus 1 revealed pathological manifestations in the upper respiratory tract," has been formally accepted for publication in PLOS Pathogens.

Best regards,

Sumita Bhaduri-McIntosh

Editor-in-Chief

PLOS Pathogens

orcid.org/0000-0003-2946-9497

Michael Malim

Editor-in-Chief

PLOS Pathogens

orcid.org/0000-0002-7699-2064